# Targeted Molecular Therapeutics for Bladder Cancer—A New Option beyond the Mixed Fortunes of Immune Checkpoint Inhibitors?

**DOI:** 10.3390/ijms21197268

**Published:** 2020-10-01

**Authors:** Olga Bednova, Jeffrey V. Leyton

**Affiliations:** 1Departément de Medécine Nucléaire et Radiobiologie, Faculté de Medécine et des Sciences de la Santé, Université de Sherbrooke, Sherbrooke, QC J1H5N4, Canada; olga.bednova@usherbrooke.ca; 2Centre d’Imagerie Moleculaire, Centre de Rechcerche, Centre Hospitalier Universitaire de Sherbrooke (CHUS), Sherbrooke, QC J1H5N4, Canada

**Keywords:** bladder cancer, antibodies, immune checkpoint inhibitors, antibody-drug conjugates, sacituzumab govitecan, enfortumab vedotin, erdafitinib, cost-effectiveness

## Abstract

The fact that there are now five immune checkpoint inhibitor (ICI) monoclonal antibodies approved since 2016 that target programmed cell death protein 1 or programmed death ligand-1 for the treatment of metastatic and refractory bladder cancer is an outstanding achievement. Although patients can display pronounced responses that extend survival when treated with ICIs, the main benefit of these drugs compared to traditional chemotherapy is that they are better tolerated and result in reduced adverse events (AEs). Unfortunately, response rates to ICI treatment are relatively low and, these drugs are expensive and have a high economic burden. As a result, their clinical efficacy/cost-value relationship is debated. Long sought after targeted molecular therapeutics have now emerged and are boasting impressive response rates in heavily pre-treated, including ICI treated, patients with metastatic bladder cancer. The antibody-drug conjugates (ADCs) enfortumab vedotin (EV) and sacituzumab govitecan (SG) have demonstrated the ability to provide objective response rates (ORRs) of 44% and 31% in patients with bladder tumor cells that express Nectin-4 and Trop-2, respectively. As a result, EV was approved by the U.S. Food and Drug Administration for the treatment of patients with advanced or metastatic bladder cancer who have previously received ICI and platinum-containing chemotherapy. SG has been granted fast track designation. The small molecule Erdafitinib was recently approved for the treatment of patients with advanced or metastatic bladder cancer with genetic alterations in fibroblast growth factor receptors that have previously been treated with a platinum-containing chemotherapy. Erdafitinib achieved an ORR of 40% in patients including a proportion who had previously received ICI therapy. In addition, these targeted drugs are sufficiently tolerated or AEs can be appropriately managed. Hence, the early performance in clinical effectiveness of these targeted drugs are substantially increased relative to ICIs. In this article, the most up to date follow-ups on treatment efficacy and AEs of the ICIs and targeted therapeutics are described. In addition, drug price and cost-effectiveness are described. For best overall value taking into account clinical effectiveness, price and cost-effectiveness, results favor avelumab and atezolizumab for ICIs. Although therapeutically promising, it is too early to determine if the described targeted therapeutics provide the best overall value as cost-effectiveness analyses have yet to be performed and long-term follow-ups are needed. Nonetheless, with the arrival of targeted molecular therapeutics and their increased effectiveness relative to ICIs, creates a potential novel paradigm based on ‘targeting’ for affecting clinical practice for metastatic bladder cancer treatment.

## 1. Introduction

Urothelial cancer typically arises from the transitional cells in the urothelium of the bladder, renal pelvis, ureter, and urethra and is commonly referred to as bladder cancer. According to the World Health Organization, bladder cancer represents the 10th most diagnosed and 13th most deadly malignancy worldwide [1]. Bladder cancer is a particular challenge to treat as it is most frequent (>50%) within the elderly and these patients often have underlying comorbidities and reduced functional status [2]. Bladder cancer is a heterogeneous disease, with 70% of cases diagnosed with superficial (non-invasive) tumors and 20% and 10% present as muscle-invasive bladder cancer (MIBC) and metastatic disease, respectively [3]. Importantly, 50–70% of superficial tumors will recur and 10–20% will progress to MIBC [3]. Despite therapy, MIBC progresses to incurable metastatic disease in about half of cases [4]. Cisplatin-containing chemotherapy regimens are the current standard-of-care for the treatment of metastatic bladder cancer. Unfortunately, the 5-year survival rate of metastatic patients treated with regimens consisting of cisplatin plus gemcitabine and methotrexate/vinblastine/doxorubicin/cisplatin is poor [5]. Chemotherapy is also related to high toxicity and has caused adverse events (AEs) of grade ≥3 in up to 82% of cases [6]. In addition, treatment related death occurs in 3–4% of cases [7]. Due to the frailty of many elderly patients, there are significant proportions of cases that are ineligible for platinum-containing chemotherapy. In these patients, carboplatin-based regimens are typically used. However, carboplatin-based chemotherapy is considered limited as the overall survival (OS) rate for these patients is approximately 9 months [5,8]. Patients who relapse after platinum-containing chemotherapy and are treated with second-line chemotherapy, have even more limited responses and poor survival [9,10]. Thus, there are unmet needs for effective and tolerable therapies for patients that are cisplatin-ineligible or for those with metastatic tumor recurrences after receiving platinum-containing chemotherapy.

In concept, precision-based therapy is the systemic administration of a drug that specifically targets tumors and, as a result, reduces nonspecific toxicities while maximizing tumor killing. Although the paradigm of targeted therapeutics has been effective and part of the standard of care for certain tumor types, it has been a challenge to accomplish in the clinic for bladder cancer [11,12,13]. The inability of targeted therapeutics to provide patients with robust patient responses or significant responses relative to chemotherapy is the major reason why chemotherapy has been the primary option for systemically treating advanced or metastatic bladder cancer for the past three decades, until the approval of immunotherapy and targeted therapeutics starting in 2016.

The advent of immunotherapies in the form of immune checkpoint inhibitors (ICIs), which are monoclonal antibodies (mAbs) that target specific factors that regulate the immune response, has dramatically changed the landscape of bladder cancer treatment. In general, ICIs work well—but only for a minority of patients providing them with long-lasting immunologic memory [14]. However, the majority of patients treated with ICIs fail to ever respond, and those that initially respond eventually develop resistance and disease progression. Unlike targeted therapeutics, there remains no predictive molecular biomarker to determine the patients who are most likely to benefit from ICI therapy.

In other tumor types, ICIs have been incredibly successful and have helped patients who have previously received not only traditional chemotherapy, but also targeted therapies [15,16,17]. A unique aspect of bladder cancer, ICIs have provided benefit in a tumor type where targeted therapy is inexistent. Bladder cancer is a unique case study because it is now targeted therapies that are coming to the rescue of patients who have previously received ICIs. Many of the ICIs have now provided results with follow-up periods that provide data with increased insight on important phase II/III trials. Thus, we provide a timely analysis of the latest clinical effectiveness of the five ICIs that are currently approved by the U.S. Food and Drug Administration (FDA). We also describe AEs, drug price, and cost-effectiveness in order to integrate important health economic insight. We found that, with some exceptions, the clinical effectiveness of ICIs is marginal and if it is worth paying a high price is justifiably questionable. In essence, the current paradigm of ICIs for bladder cancer is somewhat of a mixed fortune. 

In contrast, our review of the key trials for the targeted therapeutics enfortumab vedotin (EV), sacituzumab govitecan (SG), and erdafitinib show these agents are proving more effective than the current ICIs. EV and erdafitinib are (SG has only fast track designation) approved and now available. However, the cost of these targeted therapeutics is significantly more expensive, than the already high cost of ICIs. We also caution that these clinical results are early. This review hopes to provide clinicians and patients with the up-to-date facts in order to help decide on the best-value option for treating patients with advanced or metastatic bladder cancer. The approval process timeline is shown in Figure 1.

## 2. Up-to-Date Clinical Benefit of the Current ICI Paradigm

The five approved ICI mAbs target either programmed cell death protein 1 (PD-1) or programmed death ligand-1 (PD-L1). The rationale for targeting the PD-1/PD-L1 axis with mAbs is multi-fold. First, blocking the interaction of PD-1 and PD-L1 increases the likelihood that the immune system, if active against malignant cells, remains active. Second, levels of PD-L1 expression have been shown to correlate with bladder cancer aggressiveness and outcome. Third, the tumor mutation burden (TMB) is high, which suggests that ICIs could have significant clinical impact because of a greater T-cell-mediated antitumor immune response elicited by invasive bladder cancer [18]. Briefly, the TMB is defined as the total number of somatic mutations per DNA megabase. A thorough review on the evolution of the PD-1/PD-L1 axis in bladder cancer is reviewed in Bellmunt et al. [19].

Approvals were based on key endpoints such as objective response rate (ORR), OS and, duration of response (DOR) for locally advanced or metastatic bladder cancer. We highlight the most up-to-date clinically relevant information. AEs were also important parameters and, closely monitored and are described in a focused subsequent section.

### 2.1. Atezolizumab (Tecentriq; Genentech; South San Francisco, CA, USA)

#### 2.1.1. IMvigor210 Trial Cohort 2

On 18 May 2016, the FDA granted accelerated approval to atezolizumab, a PD-L1-targeting mAb, for use in patients with bladder cancer with locally advanced or metastatic bladder cancer who have disease progression during or following platinum-containing chemotherapy or within 12 months of receiving neoadjuvant or adjuvant platinum-containing chemotherapy [20]. Approval was based on this single-arm phase II trial. This cohort contained 310 patients with inoperable locally advanced or metastatic bladder cancer that had previously received cisplatin-containing chemotherapy [21]. Immunohistochemistry (IHC) staining evaluated the number of PD-L1-positive tumor-infiltrating immune cells (IC) and categorized patients into IC0, IC1, or IC2/3 groups. IC0, IC1, IC2, and IC3 scoring was proportional to tumors containing <1%, ≥1% and <5%, ≥5% and <10%, and ≥10% ICs within a given microscopic field of view, respectively. This PD-L1 scoring system was determined using the diagnostic assay SP142 (Ventana Medical Systems Inc. Tucson, AZ, USA). The scoring method first identifies a defined tumor area that contains at least 50 viable tumor cells [22]. For example, a score of IC3 is given when a tumor tissue shows either ≥50% of tumor cells that stain for PD-L1 or ≥10% of the tumor area is occupied with ICs that stain for PD-L1 [22]. Thus, PD-L1 scoring can be determined solely by the percentage of tumor cells or ICs that express PD-L1 and a combined score is not taken into account. The scoring for atezolizumab in patients with bladder cancer was reliant on ICs and not on tumor cells [21]. 

At the initial follow-up time of 11.7 months, the ORRs were 26% (95% CI, 18–36%) in the IC2/3 group, 18% (95% CI, 13–24%) in the IC1 and IC2/3 combined group, and 15% (95% CI, 11–19%) in all patients [21]. There were ongoing responses in 85% of responding patients and the median DOR was not reached (2.0–13.7 months). The median OS was 11.4 months (95% CI, 9.0—not reached) in the IC2/3 group, 8.8 months (95% CI, 7.1–10.6) for the combined IC1 and IC2/3 groups, and 7.9 months (95%, 6.6–9.3) in all patients. At the median follow-up of 33 months, the median ORR, OS, and DOR were 16%, 7.9 months, and 24.8 months (Table 1).

#### 2.1.2. IMvigor 210 Trial Cohort 1

On 17 April 2017, the FDA granted accelerated approval for atezolizumab in patients who are cisplatin-ineligible. This cohort consisted of 119 patients with a median age of 73 years old. The most common reason for cisplatin ineligibility was impaired kidney function. At a median follow-up time of 14.2 months the ORR was 23.5% (95% CI, 16.2–32.2%) in all treated patients [23]. Based on PD-L1 status, the ORRs were 28% (95% CI, 14–47%) and 21% (95% CI, 10–35%) for PD-L1 expression of ≥5% and <5% groups, respectively. The DOR was not reached in either subgroup. Responses were ongoing for 82% and 29% of responding patients at 5 months and 1 year, respectively. At the median follow-up of 29 months, the median ORR, OS, and DOR were 24%, 16.2 months, and not reached (95% CI: 30.4—N) (Table 1).

#### 2.1.3. IMvigor211 Trial

The phase III IMvigor211 trial compared atezolizumab with physician’s choice of chemotherapy in patients with metastatic bladder cancer who had progressed after platinum-containing chemotherapy [24]. Again, patients were stratified based on PD-L1 expression. Unfortunately, patients with the greatest relative PD-L1 expression did not significantly survive longer when treated with atezolizumab (11.1 months) relative to chemotherapy (10.6 (8.4–12.2) months) [24]. There was also no significant difference in ORR. Thus, other patient cohorts were not evaluated. The most recent results are listed in Table 1.

#### 2.1.4. IMvigor130 Trial

This randomized trial enrolled 1213 patients with locally advanced or metastatic bladder cancer who were newly diagnosed or had received neoadjuvant or adjuvant chemotherapy more than 12 months prior to commencement of atezolizumab treatment [25]. The goal was to determine the therapeutic effectiveness of atezolizumab alone or in combination with chemotherapy versus chemotherapy alone. In addition, patients were stratified by PD-L1 status as previously described. Chemotherapy was gemcitabine with cisplatin and carboplatin for cisplatin-eligible and cisplatin-ineligible patients, respectively. Although cisplatin-ineligible patients were only originally recruited, the trial was amended to include cisplatin-eligible patients. Cisplatin-ineligible and eligible patients were randomized into three treatment arms: group A—atezolizumab plus open-label chemotherapy, group B—open-label atezolizumab monotherapy, or group C—masked placebo plus open-label chemotherapy. The two primary efficacy endpoints were OS and progression-free survival (PFS). 

The most up-to-date results of the trial as reported by Galsky et al., did not statistically show that atezolizumab improved OS in all intention-to-treat patients [25]. The proportions (53–58%) of cisplatin-ineligible patients were similar among the three groups. At the median follow-up at 11.8 (6.1–17.2) months, the median OS among groups A and C were 16.0 (13.9–18.9) and 13.1 (11.7–15.1) months, respectively. This result did not cross the prespecified interim efficacy boundary for statistical significance. Group A did meet the co-primary PFS endpoint. Patients in group A had a statistically significant increased PFS of 8.2 (95% CI, 6.5–8.3) months. In contrast, patients in group C had a PFS of 6.3 (95% CI, 6.2–7.0) months. Unfortunately, the ORR for groups A and C were similar (47% (95% CI, 43–52%) and 44% (95% CI, 39–49%), respectively). 

When evaluating atezolizumab alone versus chemotherapy alone, the results were not favorable. In all patients, the OS values for atezolizumab and chemotherapy alone were 15.7 (13.1–17.8) and 13.1 (11.7–15.1) months, respectively. Although significance for OS between groups B and C were not tested, the survival curves appear to show marginal benefit with atezolizumab alone. The median ORR for group B was 23% (95% CI, 19–28%), which was much lower than the previously described ORR values for groups A and C. The PFS was not reported.

For PD-L1 subgroups, atezolizumab alone and atezolizumab plus chemotherapy improved OS relative to chemotherapy alone in patients with PD-L1 expression IC2/3. The OS for the patients treated with atezolizumab and chemotherapy was 23.6 months versus 15.9 months for chemotherapy alone. The OS for the patients treated with atezolizumab alone was not estimable (17.7—not estimable) versus 17.8 (10.0—not estimable) months for chemotherapy alone. There were no survival advantages for both groups A and B relative to C in patients with PD-L1 expression scores of IC0 or IC1. Although OS was improved ORR was not. In patients with PD-L1 IC2/3 treated with atezolizumab or chemotherapy alone were 34% (95% CI, 28–50%) and 37% (95% CI, 33–55%), respectively. 

IMvigor130 group B and IMvigor210 cohort 1 were the key trials that demonstrated a decreased response among patients with a PD-L1 status of <5% (i.e., IC0/1) when treated with atezolizumab alone, versus patients with the same PD-L1 status but who received cisplatin- or carboplatin-containing chemotherapy alone. Based on these results the FDA and the European Medicines Agency revised the indication for atezolizumab. Atezolizumab is now restricted for use in patients with locally advanced or metastatic bladder cancer who (i) are not eligible for cisplatin-containing chemotherapy, and whose tumors express PD-L1 ≥5% (based on the Ventana assay) and, (ii) are not eligible for any platinum-containing therapy regardless of PD-L1 status [37,38]. Notably, because atezolizumab did not reach its endpoints in the IMvigor211 trial, it is not indicated for cisplatin-eligible patients in the first-line regardless of PD-L1 tumor status. 

### 2.2. Pembrolizimab (Keytruda; Merck; Kenilworth, NJ, USA)

#### 2.2.1. KEYNOTE-045 Trial

On 18 May 2017, the FDA granted accelerated approval to pembrolizumab, a PD-1-targeting mAb, for use in patients with bladder cancer who have either received platinum-containing chemotherapy or who are cisplatin-ineligible [39]. The phase III KEYNOTE-045 clinical trial enrolled patients with advanced or metastatic bladder cancer previously treated with any platinum-containing chemotherapy [40]. Patients (542) were randomly assigned to receive pembrolizumab or the investigator’s choice of chemotherapy containing paclitaxel, docetaxel, or vinflunine. PD-L1 expression status was assessed using the Agilent PD-L1 IHC22C3 pharmDx assay. PD-L1 expression scores were defined as a ‘tumor proportion score’ (TPS), which is defined as the combined positive score determined from the percentage of PD-L1-expressing tumor cells and ICs relative to the total number of tumor cells. TPS values of <1%, 1–49%, ≥50% corresponding to PD-L1 expression of no expression, ‘positive’ expression, and ‘high’ expression [41].

In all patients treated with pembrolizumab the ORR was 21.1% (95% CI, 16.4–26.4%). In contrast patients treated with chemotherapy the ORR was 11.4% (95% CI, 7.9–15.8%). Pembrolizumab treatment also extended the median OS to 10.3 months (95% CI, 8.0–11.8). In comparison, the OS for patients in the chemotherapy-alone group was 7.4 (95% CI, 6.1–8.3) months. When focusing on patients whose tumor biopsies had a TPS ≥ 10%, the median OS was 8.0 (95% CI, 5.0–12.3) months compared to 5.2 (95% CI, 4.0–7.4) months for chemotherapy alone. The ORRs for the PD-L1-positive and chemotherapy alone groups were 21.6% (95% CI, 12.9–32.7%) and 6.7% (95% CI, 2.5–13.9%). Hence, there was a significant therapeutic response advantage for patients whose tumors with PD-L1 TPS ≥ 10%.

Necchi et al. and Fradet et al., recently reported the most up-to-date follow-ups of the KEYNOTE-045 trial [26,27]. The median ORR, OS, and DOR values were 21.1%, 10.1, and 29.7 months for patients treated with pembrolizumab (Table 1). In comparison, the updated median ORR, DOR, and OS values for patients in the chemotherapy group were 11%, 7.3, and 4.4 months [26]. In contrast to the earlier reported findings, patients with PD-L1 tumor TPS ≥ 10 had a lower ORR (20.3%) compared to all patients. When the effectiveness of both pembrolizumab and chemotherapy were evaluated in patients who did respond to treatment, the OS increased to 39.6 versus 17.7 months, respectively. This indicated that in patients who had been heavily pretreated with prior platinum-containing chemotherapy, have a clear increased benefit by being treated with pembrolizumab versus chemotherapy, and if responding to pembrolizumab, can survive out to >3-years. However, benefit from pembrolizumab appeared to be independent of PD-L1 expression status [40].

#### 2.2.2. KEYNOTE-052 Trial

This phase II trial was a single-arm study designed to evaluate the efficacy of pembrolizumab in patients (370) with advanced bladder cancer who were ineligible for cisplatin-containing chemotherapy [42]. The up-to-date ORR in all treated patients was 28.6% (95% CI, 24.1–33.5%) (Table 1). Responses remained ongoing in 84% of patients. Patients in the PD-L1-high expression subgroup responded better to pembrolizumab compared to the PD-L1-low subgroup. Specifically, the ORR was 47% (95% CI, 38–57%) and 21% (95%, 16–26%) in patients with PD-L1 expression TPS ≥ 10 and <10, respectively. The median DOR for both subgroups was not reached. Among all responding patients, 52% and 7% continued responding at 6 months and at 1 year. Thus, these results strengthen the use of pembrolizumab in the first-line setting for cisplatin-ineligible patients with locally advanced and unresectable or metastatic bladder cancer. 

#### 2.2.3. KEYNOTE-361 Trial

This current phase III trial (NCT02853305) is a randomized evaluation to verify the first-line effectiveness of pembrolizumab from the KEYNOTE-052 trial. Patients were stratified by their tumors either having ≥10 or <10 PD-L1 score. Unfortunately, pembrolizumab did not meet its two primary endpoints of OS or PFS [43]. As of 9 June 2020, patients receiving pembrolizumab and whose tumors were PD-L1-low had decreased survival compared to patients receiving cisplatin- or carboplatin-containing chemotherapy. We have not found any reports providing additional information on other patient cohorts.

Based on KEYNOTE-052 and KEYNOTE-361 results, the FDA issued an alert to health care professionals and oncology clinical investigators due to the substantial uncertainty concerning efficacy as a monotherapy to treat bladder cancer patients whose tumor express low (TPS < 10%) amounts of PD-L1 [44]. Cisplatin-ineligible patients should receive pembrolizumab only if their tumors express PD-L1 TPS ≥ 10. However, if patients are not eligible for any platinum containing chemotherapy, pembrolizumab can be used regardless of PD-L1 tumor status.

### 2.3. Nivolumab (Opdivo; Bristol-Myers Squibb; New York, NY, USA)

#### Checkmate 275 Trial

Nivolumab is a PD-1 blocking mAb that was approved in 2017 based on its performance in the multicenter, single-arm phase II Checkmate 275 clinical trial [45,46]. The trial evaluated Nivolumab in 270 patients with metastatic or surgically unresectable locally advanced bladder cancer, or with progression or recurrence after at least one platinum-based regimen, or within 12 months of perioperative platinum treatment for muscle-invasive disease. [46] PD-L1 expression was assessed using the Dako PD-L1 IHC 28-8 pharmDx kit [47]. PD-L1-positive staining is defined as complete and/or partial plasma membrane staining of tumor cells at any intensity. A minimum number of 100 viable tumor cells should be present in the PD-L1 stained tumor slide. Notably, infiltrating ICs that may stain positive for PD-L1 are not included in the scoring for the determination of PD-L1 positivity. 

At the median follow-up of 7 (3.0–8.8) months, nivolumab treatment resulted in an ORR of 19.6%, (95% CI, 15.0–24.9%). Regarding, PD-L1 status: PD-L1 expression scores of ≥5%, ≥1% and <5%, and <1% of tumor cells, ORR values were 24.8% (95% CI, 18.9–39.5%), 23.8% (95% CI, 16.5–32.3%) and 16.1% (95% CI, 10.5–23.1%) respectively. The median OS was 8.7 months (95% CI, 6.1 to not reached) in the overall population, 11.3 (8.7 to not reached) months in the patients with PD-L1 expression of ≥1% and 6.0 (95% CI, 4.3–8.1%) months in patients with tumors with low PD-L1 expression of <1%. DOR was not reached (7.4—NR) in all patients at the moment of the publication of these results. 

Based on these results, the FDA granted accelerated approval to durvalumab for the treatment of patients with locally advanced or metastatic bladder cancer who have disease progression during or following platinum-containing chemotherapy or have disease progression within 12 months of neoadjuvant or adjuvant treatment with platinum-based chemotherapy [45].

The most recent follow-up reported in Galsky et al., reported on the effectiveness of nivolumab [29]. At 33.7 months minimum follow-up, the ORR was 20.7% (Table 1), with complete responses in 6.7% of patients. Importantly, when the efficacy of nivolumab was evaluated in patients whose PD-L1 status was either <1% (*n* = 146) or ≥1% (*n* = 124), the ORR values were 16.4% (95% CI, 10.8–23.5%) and 25.8% (95% CI, 18.4–34.4%), respectively. When PD-L1 expression was evaluated at <5% (*n* = 187) and ≥5% (*n* = 83) the ORR values were 16.0% (95% CI, 11.1–22.1%) and 31.3% (95% CI, 21.6–42.4%), respectively. Out of the responding population, 73.2% (41/56) and 58.9% (33/56) had responses lasting ≥6- and ≥12-months, respectively. At the time of the evaluation, 25% of patients had ongoing responses. 

Importantly, TMB was an important factor for successful patient response. A ‘high’ TMB was defined as ≥170 mutations per tumor. Low and medium TMB was defined as <85 and 85–169 mutations per tumor, respectively. Cox proportional hazards regression models were used to assess the dependence of PFS and OS on TMB alone and in combination with PD-L1 scoring. The ORR was 13.0%, 19.6%, and 31.9% in patients with low, medium, and high TMBs. High TMB tumors showed a positive association with ORR (odds ratio (95% CI): 2.13 (1.26–3.60), *p* < 0.05) regardless of PD-L1 scoring. However, futher evaluation including sufficiently large numbers of patients will be needed to determine the nuances if there is a benefit in ORR with subgroups evaluating combinations of low, medium, high TMB ≥1% and <1% PD-L1. PFS and OS were longer in patients with high TMB values compared to low and medium TMB tumors. Interestingly, when TMB was analyzed in combination with PD-L1 status there were diverging trends with PFS and OS. For PFS, there was approximately a 1 month increase in patients with ≥1% PD-L1 with tumors that were both low and high TMB. However, patients with ≥1% PD-L1 had poorer OS compared to patients with <1% PD-L1 for low, medium, and high TMB. 

This study at almost 3-years of minimum follow-up indicated that nivolumab is not only effective for treating metastatic bladder cancer, but high TMB was strongly associated with improved outcomes relative to all treated patients. Moreover, TMB and PD-L1 could be used in combination as biomarkers for predicting PFS and OS. As a result this study is the first to show for ICI treatment of bladder cancer, TMB may prove a superior biomarker than PD-L1. For example, patients can be grouped into low, medium, and high TMB levels as opposed to the 1% cutoff for PD-L1. 

### 2.4. Durvalumab (Imfinzi; AstraZeneca;Cambridge, United Kingdom)

#### 1108. Trial

Durvalumab is a human mAb that binds PD-L1 and also provided encouraging results on clinical response with respect to the tumor expression status of PD-L1 and was approved in 2017. The phase I/II 1108 trial evaluated patients who had received prior platinum-based chemotherapy [48]. At the median follow-up time point of 4.3 months, the ORR was 31% (95% CI, 17.6–47.1%) in all patients. PD-L1 expression was performed using the Ventana PD-L1 (SP263) assay. The proportion of tumor cells with PD-L1 membrane staining was partitioned based on defined intervals of <1%, 1–4%, 5–9%, 10–14%, …, 90–99%, 100%. [49] Strikingly, patients in the PD-L1-positive cohort, defined as ≥25%, had an ORR of 46.4% (95% CI, 27.5–66.1%). In contrast, patients with a score of <25% were considered as PD-L1- negative cohort and had an ORR of 0% (95% CI, 0.0–23.2%).

However, updated results with a median follow-up of 5.8 (0.4–25.9) months showed that the ORR was now 17.8% (95% CI, 12.7–24.0%) in all patients [31], much lower than previously reported in the trial that led to its approval (Table 1) [48]. In addition, the ORR in the PD-L1-positive cohort dropped from 46.4% to 27.6% (95% CI, 19.0–37.5%). Nonetheless, PD-L1-positive patient ORR was still notably higher than in patients that were PD-L1-negative 5.1% (95% CI, 1.4–12.5%). The median OS was 18.2 (8.1—not estimable) months in the total population. The OS was 20.0 (11.6—not estimable) months and 8.1 (3.1—not estimable) months for the PD-L1-high and PD-L1-negative expressing cohorts, respectively. The 1-year survival rate was 55% (44–65%) in all patients. Survival rates for PD-L1-expression subgroups were 63% and 41% for PD-L1-high and PD-L1 low/negative groups respectively. Based on these first results, the FDA granted accelerated approval to durvalumab for the same indication as described for nivolumab [50]. This study was significant as, for the first time, the results indicated that patient tumors should most likely contain substantially high levels of PD-L1. PD-L1 expression cut-offs at 1–10% may be insufficient for identifying patients who will respond to ICI therapy. 

Durvalumab is currently being investigated in combination with the ICI mAb tremelimumab that targets CTLA-4 (another immune checkpoint receptor) in a few different clinical trials and is reviewed in [51]. Unfortunately, in one significant trial, the phase III DANUBE trial evaluating durvalumab plus tremelimumab in unresectable, metastatic bladder cancer patients did not meet the primary endpoints for improving OS versus standard-of-care chemotherapy [52]. The trial is evaluating the efficacy of durvalumab in the first-line treatment of both cisplatin-eligible and -ineligible patients with metastatic bladder cancer. The trial arms are durvalumab monotherapy, durvalumab plus tremelimumab, and cisplatin and gemcitabine or carboplatin and gemcitabine chemotherapy. In addition, patients whose tumors were PD-L1-positive did not benefit from durvalumab plus tremelimumab. This was surprising since high PD-L1 was set at the high cut-off of ≥25% of only tumor cells expressing PD-L1. Taken together, these results are puzzling. On one hand, durvalumab effectiveness is associated with PD-L1 expression as a monotherapy in the second-line but not in combination with an additional non-overlapping ICI in patients with metastatic bladder cancer in a first-line setting. The DANUBE trial is a post-approval commitment from AstraZeneca in agreement with the FDA from the accelerated 2017 approval, and it is unclear what actions regarding its approval and/or use will follow. Result details have yet to be published or presented at the time of this review. 

### 2.5. Avelumab (Bavencio; Pfizer; New York, NY, USA)

#### 2.5.1. JAVELIN Solid Tumor Trial

Avelumab is another fully human mAb that targets PD-L1. In the phase Ib JAVELIN clinical trial [33], avelumab was studied in 249 patients with metastatic bladder cancer previously treated with platinum-containing chemotherapy. PD-L1 expression was assessed using the Dako PD-L1 IHC 28-8 pharm Dx assay. The scoring system was similar to the Dako 28-8 assay used for nivolumab. Notably, scoring only accounted for PD-L1 expression on tumor cells only. At 6 months follow-up, the ORR was 17% (95% CI, 11–24%). In PD-L1-positive (≥5% of tumor cells) patients, the ORR was 24% (95% CI, 14–36%). In patients whose tumors had a PD-L1 status of <5% tumor cells, the ORR was 13% (95% CI, 7–23%). The median DOR for all patients was not reached (95% CI, 42.1 weeks to not estimable). Median OS was 6.5 (95% CI, 4.8–9.5) months in all patients. Patients in the PD-L1 status ≥5% and <5% subgroups, the median OS values were 11.9 (6.1–18.0) and 6.1 (5.9–8.0) months, respectively. 

#### 2.5.2. JAVELIN Bladder 100 Trial

This was followed with the phase III JAVELIN Bladder 100 trial that evaluated 700 patients given gemcitabine with either first-line cisplatin or carboplatin with and without maintenance avelumab plus best supportive care (BSC; *n* = 350) or BSC alone (*n* = 350) [32]. At the median follow-up time of approximately 19 months, avelumab plus BSC significantly prolonged OS versus BSC alone. The median OS for avelumab plus BSC was 21.4 months compared to 14.3 months for BSC alone. Of note, patients with PD-L1-positive tumors had a median OS that was not reached versus 17.1 months for BSC alone. The ORR of 17% and 24.1% in all patients and PD-L1 ≥5% ICs, respectively, obtained from the phase Ib study is the only available ORR (Table 1) as no ORR for the phase III study has been reported as of submission of our findings. Nonetheless, avelumab has thus far reached its primary objective in a large-scale randomized trial. Based on these results, the FDA approved avelumab on June 20, 2020 for the treatment of patients with locally advanced or metastatic bladder cancer that has not progressed with first-line platinum-containing chemotherapy [53]. 

Avelumab is currently being studied in the GCISAVE trial (NCT03324282) that will assess the effectiveness of avelumab in combination with gemcitabine/cisplatin in the first-line treatment of locally advanced metastatic bladder cancer. Avelumab is also currently being evaluated in combination with Bacille Calmette-Guerin in patients with non-muscle invasive bladder cancer (NCT03892642), radiation (NCT03747419), and KHK2455 (a indoleamine 2,3-dioxygenase inhibitor; NCT03915405) in patients with advanced bladder cancer. These findings indicate that avelumab is very promising as maintenance therapy in patients who respond after receiving first-line platinum-containing chemotherapy. In addition, we look forward to ORR results from the Bladder 100 trial and the results of the multiple combination strategies currently being evaluated in the clinic. 

### 2.6. Comparative Nuances between Studies for PD-L1 Expression as a Biomarker

Atezolizumab: PD-L1 expression scoring was based solely on ICs and not tumor cells. In patients who previously received platinum-containing chemotherapy, the IMvigor210 (cohort 2) trial showed patients achieved ORRs of 26% in the IC2/3 (≥5%) group compared to 18% in the IC0/1 group. Unfortunately, in the IMvigor211 trial patients with the greatest relative PD-L1 expression did not significantly survive longer when treated with atezolizumab. However, the Imvigor210 (cohort 1) patients with who were cisplatin-ineligible, the ORRs were 28% and 21% for PD-L1 expression of ≥5% and <5% groups, respectively. The IMvigor130 trial did show an improved OS in patients with PD-L1 IC2/3. However, patients with PD-L1 IC2/3 actually had poorer relative ORR values.Pembrolizumab: PD-L1 expression scoring was based on a combination of both tumor cells and ICs. Although initial reports from the KEYNOTE-045 trial demonstrated increased ORR for patients with high PD-L1 expression, longer follow-up reports did not show an outcome advantage for patients with high expression levels of PD-L1. In cisplatin-ineligible patients (KEYNOTE-052), there was an association with PD-L1 expression and patient outcome. However, the phase III KEYNOTE-361 trial showed that patients in the PD-L1 high group did not have improved PFS or OS.Nivolumab: PD-L1 expression scoring was based solely on tumor cells and not ICs. Although clear differences in ORR were observed when patients were grouped into ≥1% (25.8%) versus ≥5% (31.3%) PD-L1 expression, TMB provided the best predictor of response. An additional complication is when TMB was combined with <1% or ≥1% PD-L1 expression, OS was dramatically reduced indicating PD-L1 was a negative predictor in this context.Durvalumab: PD-L1 expression scoring was based solely on tumor cells and not ICs. The 1108 trial has demonstrated differences in ORR and OS based on PD-L1 expression levels. One potential explanation is the much higher expression threshold of 25%. ORR and OS were 27.6% and 20.0 months for patients with ≥25% PD-L1 expression. In contrast, ORR and OS were 5.1% and 8.1 months for patients with <25% PD-L1 expression. This indicates, that much higher levels of PD-L1 expression cutoffs may provide improved prediction of patient outcomes.Avelumab: PD-L1 expression scoring was based solely on tumor cells and not ICs. At a threshold of 5%, patients in the ≥5% group had ORR and OS values of 24% and 11.9 months, respectively. In contrast, the ORR and OS for patients in the <5% group were 13% and 6.1 months, respectively.

## 3. Antibody-Drug Conjugates

Antibody-drug conjugates (ADCs) are the most mature offshoot of unmodified mAb therapeutics [54]. ADCs are mAbs conjugated to a small molecule chemotherapeutic via a chemical crosslinker. Although ADCs are considered biological therapeutic agents, the concept can also be viewed as ‘targeted chemotherapy’. ADC therapeutic efficacy is reliant on the ability to efficiently internalize and accumulate the delivered cytotoxic drug inside diseased cells. Cells constantly internalize extracellular ligands via receptor-mediated endocytosis. Often, these internalized ligand-receptor complexes are encapsulated inside endosomes and trafficked to lysosomes for enzymatic degradation. Mechanistically, ADCs exert their cytotoxic activity by binding to target antigen receptors on the surface of tumor cells where they are internalized by a process known as receptor-mediated internalization and are entrapped inside endosomes in the intracellular space. Motor proteins then naturally traffic endosomes to lysosomes for membrane fusion and transfer of the encapsulated contents. Lysosomal proteases digest the antibody backbone or cleave the chemical crosslinker and liberate functional chemotherapeutic metabolites. The metabolites are able to permeate the lysosomal membrane and diffuse and bind their target and inhibit their function [54]. We highlight two recent clinically successful ADCs for bladder cancer.

### 3.1. Enfortumab Vedotin (Padcev; Astellas; Tokyo, Japan; and Seattle Genetics; Bothell, WA, USA)

EV is an ADC that targets Nectin-4 that is overexpressed on the surface of bladder tumor cells. EV is conjugated to the microtubule inhibitor monomethyl auristatin E that causes G2/M cell cycle arrest and results in apoptosis. [55] Nectins are involved in cellular adhesion, migrations and polarization [56]. IHC analysis showed Nectin-4 to be overexpressed in 93% of metastatic urothelial tumor specimens [55]. In contrast, 294 normal tissue specimens representing 36 human organs showed homogeneous weak to moderate staining. Nectin-4 expression via IHC staining was determined by the H-score. The H-score was calculated by summing the products of the staining intensity (score of 0–3) multiplied by the percentage of cells (0–100) stained in a given field of tumor tissue [55]. Specimens were then classified as negative (H-score 0–14), weak (H-score 15–99), moderate (H-score 100–199), and strong (H-score 200–300). Thus, Nectin-4 is an attractive target due to its preferential overexpression in bladder cancer relative to normal tissues.

#### 3.1.1. EV-101 Trial 

EV was approved in the United States in December 2019 based on the results from the phase I and II EV-101 and EV-201 clinical trials, respectively (extensively reported in [57]) (Table 1) [58]. Of note, patients in the EV-101 study who had previously received ICI therapy had an ORR of 42% (95% CI, 31.2–52.5%). In addition, patients with high tumor burdens such as liver metastases had a 36% (95% CI, 20.4–54.9%) ORR [59].

#### 3.1.2. EV-201 Trial 

In the phase II EV-201 trial, 125 patients with locally advanced or metastatic bladder cancer who were previously treated with platinum-containing chemotherapy or ICI therapy were treated with EV. Tumor expression levels of Nectin-4 and PD-L1 were evaluated. Nectin-4 expression levels were evaluated and scored as previously described [55]. The median Nectin-4 expression level was H-score = 290 (14–300) and, hence, all patient tumors evaluated were positive for Nectin-4 and, they were considered as having ‘strong’ expression. PD-L1 expression was scored as previously performed for pembrolizumab with tumors being classified as positive with a score ≥10 [42]. The proportion of patients with <10 and ≥10 PD-L1 scores was 65% and 35%, respectively.

At the median follow-up time point of 10.2 (0.5–16.5) months the ORR was 44% (95% CI, 35.1–53.2%), including 12% with complete responses regardless of PD-L1 status (Table 1) [34]. The median DOR was 7.6 (4.9–7.5) months. The median OS was 11.7 (9.1—not reached) months. Hence, these patient outcomes are reflective for tumors that had ‘strong’ Nectin-4 expression. In contrast, in the PD-L1 subgroups the ORRs were 47% (95% CI, 36–59.1%) and 36% (95% CI, 21.6–52%) for TPS < 10 and TPS ≥ 10 PD-L1 expression scores, respectively. This indicated that patients responded regardless of PD-L1 expression. This study demonstrated, (i) the importance of Nectin-4 as a biological target for bladder cancer, relative to PD-L1, and (ii) EV has the potential to significantly extend the lives of patients, including those who failed ICI treatment.

EV is currently being evaluated in the phase III EV-301 clinical trial (NCT02091999). In this global study, approximately 550 patients are being randomized to receive EV or investigator’s choice of docetaxel, paclitaxel, or vinflunine [60]. There is also currently active recruitment for a phase II study (NCT03288545) to evaluate EV alone and in combination with various anticancer therapies, including the ICI pembrolizumab [61,62]. Preliminary results have shown an ORR of 71% with the combination of EV plus pembrolizumab in 45 cisplatin-ineligible patients. The available data for DOR, PFS, and OS are not yet mature. Taken together, these clinical studies reveal EV has the potential to significantly extend the lives of patients who fail ICI treatment and has the potential to synergize patient response with ICI therapy.

### 3.2. Sacituzumab Govitecan (Trodelvy; Immunomedics; Morris Plains, NJ, USA)

SG is an ADC that targets Trop-2 that is overexpressed on the surface of bladder tumor cells. SG is conjugated to the topoisomerase I inhibitor SN-38 [63]. It is currently approved for use in patients with triple-negative breast cancer who have received at least two prior forms of chemotherapy. Trop-2 is a transmembrane glycoprotein, which participates in cellular self-renewal, invasion, proliferation and survival and overexpressed in multiple solid tumors, including bladder cancer [64]. An IHC analysis showed Trop-2 was generally overexpressed in bladder tumor tissue with little expression detected in the corresponding normal tissue [65]. The method in which Trop-2 expression level was determined was not provided. 

#### IMMU-132 Trial

SG was evaluated in the phase I/II IMMU-132 clinical trial in patients with advanced bladder cancer that received prior platinum-based treatment (Table 1) [35]. Patient tumors were determined as positive if >10% of tumor cells had anti-Trop-2 staining. Expression was scored as 3+ (strong), 2+ (moderate), and 1+ (weak) [35]. Tumors with <10% tumor cells that stained for Trop-2 were considered Trop-2-negative tumors. SG treatment resulted in an ORR of 31%, with two complete and 12 partial responses out of 45 patients. In a patient cohort previously treated with ICIs, the ORR was 23% (4/17). A single-arm, open-label, global TROPHY U-01 phase II trial is currently ongoing to evaluate SG in advanced bladder cancer (NCT03547973). Interim results from 35 patients from the 100-patient cohort of cisplatin-eligible patients who have also previously received ICI therapy and platinum-containing chemotherapy showed an ORR of 28% [66]. Based on these results, the FDA granted Immunomedics request for fast track designation in order to make SG available as rapidly as possible. It is not known whether there was a correlation in patient responses with respect to Trop-2 expression levels.

## 4. Erdafitinib (Balversa; Janssen Pharmaceuticals; Beerse, Belgium)

Bladder cancer is the third most common mutated malignancy and has the strongest association to fibroblast growth factor receptors (FGFRs) 1–4 gene mutations relative to all other cancer types [67,68]. Moreover, FGFR mutational aberrations occur in >50% of all bladder cancer cases [68]. Interestingly, FGFR3 mutations occur in 60% of invasive bladder tumors and it is a poor prognostic marker [69].

FGFRs represent a family of tyrosine kinases found on the surface of normal cells. There are currently four recognized receptor isoforms, which bind corresponding ligands, and leads to receptor dimerization and phosphorylation [68]. Ligand binding and dimerization results in downstream signaling and expression of several gene products that function to promote cell survival and proliferation. Gene mutations including gene amplification combine to promote cell growth beyond normal limits and results in the development of cancer. Further abnormal FGFR mechanisms that promote bladder cancer is nicely reviewed in Roubal et al. [70].

### BLC2001 Trial

Erdafitinib is a pan-FGFR inhibitor and exerts is action by binding to and blocking FGFR phosphorylation and signaling, which decreases cell viability, particularly in tumor cells with FGFR genetic alterations. It was approved on 12 April 2019 for use in patients with locally advanced or metastatic bladder cancer, with susceptible FGFR3 or FGFR2 genetic alterations, that has progressed during or following platinum-containing chemotherapy, including within 12 months of neoadjuvant or adjuvant platinum-containing chemotherapy [71]. Approval was based on its performance in the phase II BLC2001 clinical trial (Table 1) [36]. Patients with locally advanced and unresectable or metastatic bladder cancer, with prespecified FGFR alterations, and who had been previously treated with platinum-containing chemotherapy were enrolled into the study. Importantly, a proportion of enrolled patients also received prior ICI therapy. Erdafitinib treatment resulted in an impressive ORR of 40% (95% CI, 31–50%). The median OS was 13.8 (9.8—not reached) months. In addition, at 1 year, 19% (11–29%) of patients continued to respond to treatment. Interestingly, Patients with FGFR3 mutations had the highest ORR at 49%. In contrast, patients with FGFR fusions had the lowest ORR at 16%. This indicates that erdafitinib can serve patients more effectively with tumors that contain FGFR mutations as opposed to fusions. In addition, erdafitinib can improve the outcomes of patients who previously received ICI therapy.

These targeted drugs, finally, are clinical breakthroughs that demonstrate molecular targets can result in precision therapeutics that are highly effective against metastatic bladder cancer. In addition, these drugs can also improve outcomes of patients that don’t respond or relapse after receiving ICI therapy. 

## 5. Adverse Events

Comparative percentages between the described ICIs and targeted therapeutics for % any AE, % grade ≥3 AE, % discontinued due to AE, and % treatment related deaths are listed in Table 2. The median % any AE was 64% (60.7–69.3%) among the key ICI clinical trials described in this review. For the ICIs, fatigue was the most commonly observed AE. Other observed AEs specifically related to ICI therapy were asthenia, infusion-related reactions, diarrhea, anorexia, peripheral edema, pruritus and rash. Severe AEs of grade ≥3 were fatigue, anemia, hepatitis, increased lipase and amylase, diarrhea, and asthenia. Grade 5 treatment-related pneumonitis that resulted in death, occurred in patients treated with durvalumab, nivolumab, and avelumab [31,33,46]. Nivolumab caused a death due to respiratory failure. Durvalumab caused a death due to autoimmune hepatitis. Pembrolizumab caused death due to sepsis and myositis [28].

For EV, SG, and erdafitinib, the most common treatment-related AEs was also fatigue. Other AEs common for the targeted therapeutics were diarrhea, nausea, any peripheral neuropathy, neutropenia, alopecia, any rash, decreased appetite and dysgeusia, hyperphosphatemia, and stomatitis. Severe AEs of grade ≥3 were neutropenia, anemia, hypophosphatemia, fatigue, leukopenia, hyponatremia, stomatitis, and asthenia. The most common reason for treatment discontinuation were retinal pigment epithelium, hand-foot syndrome, dry mouth, and skin or nail events [35,36,72]. Notably, thus far there have been no treatment-related deaths. However, a note of caution is that there were increased proportions of patients that discontinued treatment relative to ICIs. The median % discontinued due to AE was 6% (1.6–9.2%). In contrast, EV, SG and erdafitinib had % discontinued due to AE of 1.5–2.2-fold higher. In addition, the % grade ≥3 AE category the targeted therapeutics was increased by a factor of 2.6–3.4. Although the targeted therapeutics appear to be sufficiently tolerated or AEs are appropriately managed, they are more toxic than ICIs and thus the safety of patients should be closely monitored.

## 6. Health Economic Factors

The innovative therapeutic approach brought by ICIs has undoubtedly ushered a new paradigm for treating patients with metastatic bladder cancer. ICIs have provided physicians the ability to control tumor growth, extend survival, and can be administered with a better safety profile compared to traditional chemotherapy. This is a major advancement for a disease that afflicts patients that are typically elderly and are frail or have co-morbidities, and cannot tolerate harsh chemotherapy. However, the exorbitant cost of these ICIs combined with the latest follow-up results (Table 1) demonstrating either less than hoped for patient responses or failure to meet endpoints in critical phase III trials that were a condition of their accelerated approval, their high economic burden and cost-effectiveness is now widely debated [74,75,76]. 

### 6.1. Drug Costs

Table 3 shows the current price ($US)/mg of ICI, a typical dose, and the price/dose. The current attitude is that these drugs cost too much [74]. Renner et al., has pointed out that ICIs may reduce toxicity but they are financially toxic and high costs limit their access in many countries inside and outside the U.S. [77]. A potential drawback for targeted therapeutics, the ADCs EV and SG and, the small molecule erdafitinib are more expensive than the listed ICIs. Unlike ICIs, ADCs are composed of three key components (antibody, chemical crosslinker, and cytotoxic payload). ADC construction involves chemical conjugation steps that can be complicated and make production and purification difficult. 

### 6.2. Cost-Effectiveness

#### 6.2.1. Pembrolizumab for Patients Who Have Progressed within 12 Months of Neoadjuvant or Adjuvant Platinum-Containing Chemotherapy Regardless of PD-L1 Expression

The National Comprehensive Cancer Network and the European Society for Medical Oncology recommend pembrolizumab in their treatment guidelines for patients who relapse after any platinum-containing chemotherapy [18,78]. This was due to the evidence from the KEYNOTE-045 trial and pembrolizumab is the only one of the five approved ICIs to demonstrate increased survival compared to standard chemotherapy after progression on platinum-containing chemotherapy. 

However, there appears to be uncertainty in the cost-effectiveness of pembrolizumab. A 2018 analysis by Sarfaty et al., based off data from KEYNOTE-045 determined that for this indication relative to chemotherapy pembrolizumab did not produce the quality-adjusted life years (QALY) gains at a willingness-to-pay threshold of $100,000 in the U.S. [79]. QALY is a measure of the incremental health improvement provided by a new treatment compared to previous treatment options. The cost-effectiveness ratio for pembrolizumab was calculated at $122,557/QALY in the U.S. The author’s did find that pembrolizumab was cost-effective in Canada, Australia, and the United Kingdom (UK), as the costs were below the $100,000 threshold. This finding was reliant on short-term data obtained from the trial. In comparison, Slater et al. analyzed results from the KEYNOTE-045 study at a median follow-up of >2-years [80]. The study reported that at a willingness-to-pay threshold of $100,000, pembrolizumab is a cost-effective option ($93,481/QALY gained) compared to chemotherapy in the U.S.

Unlike in the U.S., many countries have cost-effectiveness assessment agencies. In March 2020, the UKs National Institute for Health and Care Excellence (NICE) Evidence Review Group (ERG) recommended against the use of pembrolizumab for this indication based on cost-effectiveness estimates (Table 3) [81]. The current list price for pembrolizumab in the UK is £26.30/mg [82]. As a single treatment course is 200 mg this amounts to approximately £5260 for a single administration. Administration of pembrolizumab is recommended at 200 mg each 3 weeks until disease progression, unacceptable toxicity, or up to 24 months without disease progression. NICE projected that pembrolizumab will cost well beyond £50,000/QALY, which is NICE’s limit threshold for a drug that can extend life ≥3 months in patients suffering from a disease with a life expectancy of <2-years. We believe, these results reveal that cost-effectiveness for pembrolizumab is difficult to justify. 

#### 6.2.2. Pembrolizumab for Patients Who Are Cisplatin-Ineligible

NICE’s ERG did recommend the use of pembrolizumab for use in patients who are cisplatin-ineligible. NICE projected the cost-effectiveness of pembrolizumab for this indication will be £67,068/QALY ($U.S. ~87,000) [83]. Merck Sharp & Dohme economic modeling projected a cost of £37,081/QALY ($U.S. ~49,000). NICE did acknowledge that there was a level of uncertainty in the calculated cost-effectiveness projections as the data was from the KEYNOTE-052 phase II study and not a randomized phase III study. This meant that the extrapolation of OS and PFS in patients treated with pembrolizumab were compared to an independent comparator arm that received gemcitabine plus carboplatin reported by De Santis et al. [8]. In the U.S., Merck Sharp & Dohme calculated that the cost-effectiveness of pembrolizumab in this setting will be $81,493/QALY at a willingness-to-pay threshold of $100,000. [84] However, it is unknown if the cost-effectiveness in the UK or the U.S. of pembrolizumab has changed since the announcement that it did not reach its primary endpoints in the critical phase III KEYNOTE-361 trial [43]. Thus, we believe the cost-effectiveness for pembrolizumab is currently difficult to justify and will most likely be deemed not cost-effective.

#### 6.2.3. Atezolizumab for Patients Who Have Progressed within 12 Months of Neoadjuvant or Adjuvant Platinum-Containing Chemotherapy Regardless of PD-L1 Expression

As pembrolizumab and atezolizumab are the only ICIs evaluated in randomized controlled trial for this bladder cancer treatment setting, Slater et al., performed a cost-effective evaluation comparative study [80]. The analysis compared a >2-year follow-up for pembrolizumab’s KEYNOTE-045 trial with the data from atezolizumab’s IMvigor211 trial. The study found because atezolizumab was less effective at extending the lives of patients, it use would increase costs by $26,458 in the U.S. for the same QALY-gained with pembrolizumab. Pembrolizumab had a cost-effective ratio of $93,481/QALY-gained. Thus, at a willingness-to-pay threshold of $100,000, the increased costs for using atezolizumab make it non cost-effective option for treating bladder cancer for this indication. The Scottish Medicines Consortium also determined that atezolizumab was not cost-effective for use within its healthcare system [85]. The primary reason was the small numerical increase in median OS compared with chemotherapy. 

#### 6.2.4. Atezolizumab for Patients Who Are Cisplatin-Ineligible

NICE’s ERG performed an analysis of the cost-effectiveness of atezolizumab for cisplatin-ineligible patients. NICE recommended atezolizumab as an option for untreated advanced or metastatic bladder cancer in patients who are ineligible for cisplatin-containing chemotherapy [86]. However, the ERG did note that although atezolizumab appears to be an effective treatment, it is difficult to establish the size of the clinical benefit compared with current treatments. Clinicians invited to comment during the review commented that atezolizumab therapy was not favorable over current treatments. The ERG further notes that they are awaiting data from the IMvigor130 trial. The trial has shown that the addition of atezolizumab to chemotherapy was associated with a significant prolongation of PFS and able to improve OS particularly in patients with a relative high PD-L1 expression status. NICE has not updated its findings or recommendation. We believe atezolizumab is likely cost-effective in this setting for patients with high PD-L1 status.

#### 6.2.5. Avelumab

Avelumab has been studied and deemed that its use would have a cost-neutral impact within a US commercial and a Medicare health plan [87]. Based on that it is the lowest priced drug on the list (Table 3), and the positive data from the phase III JAVELIN Bladder 100 trial, avelumab is a relatively affordable and a cost-effective option.

#### 6.2.6. Nivolumab and Durvalumab

NICE did not recommend nivolumab as an option for treating locally advanced, unresectable or metastatic bladder cancer who have had platinum-containing therapy [88]. The cost-effectiveness was not better than cisplatin plus gemcitabine. Nivolumab was also shown to not be cost-effective in the US economic healthcare system [76]. This may be due to nivolumab having the highest relative toxicity profile among the ICIs. For durvalumab, AstraZeneca advised that they would not be pursuing a licensing application from the European Medicines Agency for a bladder cancer indication and, thus, NICE has suspended its cost-effectiveness appraisal [89]. 

#### 6.2.7. ADCs and Erdafitinib

There are currently no cost-effective appraisals for EV or for SG (for its current approval for use in triple-negative breast cancer). For erdafitinib, NICE is currently appraising its cost-effectiveness and results are to come [90].

## 7. Discussion

Patients with metastatic or advanced bladder cancer once had limited options after failed chemotherapy leading to disease progression and death. Active application and examination of immune checkpoint inhibition has provided new therapeutic possibilities for patients with metastatic bladder cancer. Today, patients typically receiving ICI therapy do not have to withstand the severe toxicity associated with chemotherapy. In addition, patients who respond typically have long-lasting responses and increased survival. However, the median ORR from the clinical trials evaluating the described ICIs was 20.9% (13.4–28.6%). In contrast for the two approved targeted therapeutics, EV and erdafitinib, the ORR values were increased by factors of 2.1 and 1.9, respectively. In addition, ICI affordability, not only for patients, but also for national health care systems threatens patient access to these drugs. However, the targeted therapeutics are even more expensive than the ICIs. The findings that avelumab and atezolizumab (for cisplatin-ineligible patients only) are most likely the only cost-effective ICIs provides a wake-up call to develop strategies to make these drugs more affordable and/or how to improve patient responses. 

As described in this review, in general, it is difficult for physicians to identify patient groups that will benefit from ICI therapy based on PD-L1 tumor expression and has previously been discussed [91]. Some patients have demonstrated strong responses under ICI therapy whose tumors express relative ‘high’ levels of PD-L1, such as in the JAVELIN Solid Tumor (avelumab) and in the Checkmate 275 (nivolumab) trials. However, in general PD-L1-specific responses were not better than in patients regardless of PD-L1 expression, or PD-L1 associated responses were not reproducible in larger randomized trials or when responses were evaluated at longer follow-up periods. The major reason for these scattered results, in the context of PD-L1 expression, is that PD-L1 is an unreliable marker to predict treatment response. One major challenge for PD-L1 as a biomarker is the different assays and expression scoring systems used, as described in the above ICI clinical trials. Currently, different IHC assays have different PD-L1 expression cutoffs and scoring is either only on tumor cells (nivolumab, durvalumab, avelumab), only on tumor-infiltrating ICs (atezolizumab), or on the combination of tumor cells and ICs (pembrolizumab). Attempts to standardize PD-L1 expression evaluation using IHC are underway. Preliminary harmonization studies have indicated that the assays 22C3, SP263 and, 28–8 (used for pembrolizumab, durvalumab and avelumab, respectively) can be comparable, additional research is needed regarding the interchangeability of the assays as it pertains to response and once a universal assays is in place, what will be the PD-L1 expression thresholds required to achieve robust responses [92].

ADCs and small molecules have given another life-saving chance to patients with advanced and metastatic bladder cancer, and have demonstrated a higher ORR in clinical trials, compared to ICI therapies, but increased frequency of any-grade treatment-related AE (Table 2) and high cost (Table 3) remain a serious barrier for mainstream application in patients. In addition, these results are early and longer follow-up analyses are needed. Nonetheless, EV and erdafitinib are effective for treating bladder cancer and, hence, inaugurated the era of targeted therapy for bladder cancer.

## 8. Future

### 8.1. Improved Biomarkers

Biomarkers are needed that will be able to identify patients for treatment-specific responses. Unlike, other tumor types, there remains no biomarker in the clinic that allows physicians to determine which patients are most likely to benefit from immuno- or targeted therapeutics. As described in the Checkmate 275 trial, TMB appears to be a promising biomarker. There are active investigations for developing bladder cancer-specific biomarkers and reviewed in [12,93]. 

The antigens Nectin-4 and Trop-2 as biomarkers to identify patients to respond to EV and SG, respectively, appear promising. Specifically, the fact that all patients were positive for Nectin-4 and that the majority of these patients had ‘strong’ expression is highly encouraging. Although, the EV-201 trial performed many subgroup analyses, it did not report patient responses based on ‘low’, ‘moderate’, and ‘high’ Nectin-4 expression. [34] This was most likely because the majority of patients had ‘high’ Nectin-4 expression and there may have been too few patients with ‘low’ and ‘moderate’ expression. A larger phase II EV-202 trial (NCT04225117) is currently recruiting and estimates to enrol 240 patients and perhaps they will determine patient responses based on Nectin-4 tumor expression.

### 8.2. Additional Targeted Therapeutics in the Pipeline

Emerging targeted therapies that have reached the clinic include inhibitors against angiogenesis, FGFR, HER2, phosphoinositide 3 kinase, protein kinase B, mammalian target of rapamycin, and epigenetic targets and are nicely described by Mendiratta and Grivas [12]. Notably, many of the investigational drugs have not shown significant activity reinforcing the difficulty with the targeted therapy approach for bladder cancer. 

MAbs make up a large portion of these investigational drugs. For example, mAbs such as bevacizumab and ramucirumab that target vascular endothelial growth factor (VEGF) have been evaluated. Bevacizumab failed to improve OS relative to placebo in a phase III study, and caused grade ≥3 AEs in 83.4% of patients [94]. In the RANGE phase III trial, ramucirumab did not significantly improve OS in patients who had previously been treated with platinum-containing chemotherapy and/or ICI therapy [95]. The mAb trastuzumab that targets HER2 has also been extensively evaluated in the clinic against bladder cancer. Unfortunately, trastuzumab has also not had any significant clinical impact. One of the great examples of antibody-targeted therapy for cancer is the story of HER2 [54]. Trastuzumab in combination with paclitaxel is standard practice for patients with HER2-positive breast cancer. Initially, a phase II study showed a remarkable ORR of 70% in patients with advanced bladder cancer treated with trastuzumab plus chemotherapy [96]. However, a larger trial that evaluated chemotherapy with and without trastuzumab did not show a difference between the two arms for bladder cancer patients [97]. 

ADCs may be the best option for antibody-based therapies relative to unmodified mAbs. The ICIs and the above described mAbs targeting VEGF and HER2, and mAbs in general, that are reliant on an antagonistic (blockade or receptor-ligand or receptor-receptor interactions) mechanism of action may show some therapeutic potency—the effects tend to be various and ultimately not curative [98]. The strategy of conjugating chemotherapy drugs to mAbs to generate ADCs appears to be the more clinically successful approach for antibody-based treatment of metastatic bladder cancer. Hence, future research directions should discover additional antigens that are overexpressed on the surface of bladder cancer, and for the development of ADCs that deliver highly cytotoxic payloads. One example is the discovery of the interleukin-5 receptor α-subunit (CD125) as being preferentially overexpressed in MIBC tumors but not on superficial bladder tumors or normal urothelium [99]. An anti-CD125 ADC has potent cytotoxicity against MIBC cells [99]. Additional target with accompanying ADCs that have shown promise in preclinical models of bladder cancer include The Slit- and Trk-like receptor family, transmembrane glycoprotein epithelial cell-adhesion molecule, and Thomsen-Fridenreich antigen and are described in further detail in [100].

## 9. Conclusions

ICIs have greatly reduced AEs compared to traditional chemotherapy. However, their relatively low response rates make it unclear on their ability to increase the therapeutic window relative to traditional chemotherapy remains unclear. In addition, their high cost makes ICIs, with the exception of avelumab and atezolizumab (for cisplatin-ineligible patients), not cost-effective. Based on the evidence described in this review, newly diagnosed patients with advanced bladder cancer will most likely significantly benefit from avelumab plus cisplatin-containing or carboplatin-containing chemotherapy. The targeted therapeutics EV, SG, and erdafitinib still have to demonstrate their worth in randomized phase III testing. If successful, it is likely patients who have relapsed after traditional chemotherapy or ICI therapy will benefit from these targeted therapeutics. Bladder cancer therapy has advanced tremendously in a short period of time since the first ICI approval, and the future looks hopeful as science will increase knowledge to make responses more robust to ICI therapy or new targetable biomarkers will be discovered.

## Figures and Tables

**Figure 1 ijms-21-07268-f001:**
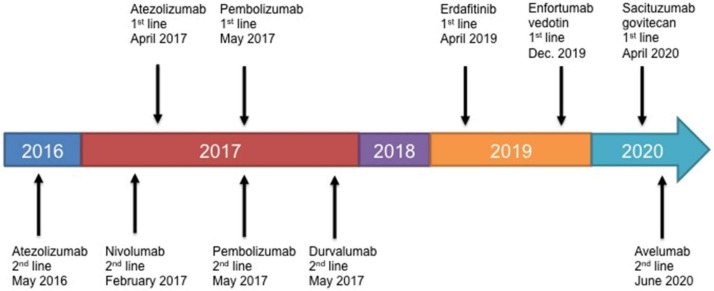
FDA approval timeline for ICIs, EV, and Erdafitinib against bladder cancer.

**Table 1 ijms-21-07268-t001:** Updated clinical trial ORR, OS, and DOR for ICIs and targeted agents as monotherapies.

ICIs	Trial and Updated (Ref)	ORR (95% CI)	OS (95% CI; Months)	DOR (95% CI; Months)
Atezolizumab ^1^	IMvigor210 Phase II [23] June 2018	Cohort 1: 24% Cohort 2: 16%	16.2 (10.4–24.5) 7.9 (6.7–9.3)	NR (30.4-NR) 24.8 (13.8–30.4)
Atezolizumab ^2^	IMvigor211 Phase III [24] December 2017	13.4%	8.6 (7.8–9.6)	21.7 (13.0–21.7)
Atezolizumab ^3^	IMvigor130 Phase III [25] May 2020	23%	15.7 (13.1–17.8)	NR (15.9-NR)
Pembrolizumab ^1^	KEYNOTE-045 Phase III [26,27] September 2019	21.1%	10.1 (8.0–12.3)	29.7 (1.6–42.7)
Pembrolizumab ^2^	KEYNOTE-052 Phase II [28] June 2020	28.6%	11.3 (9.7–13.1)	30.1 (18.1-NR)
Nivolumab	CheckMate 275 Phase II [29] June 2020	20.7%	8.6 (6.1–11.3)	20.3 (11.5–31.3)
Durvalumab	Study 1108 Phase I/II [30,31] July 2018	17.8%	10.5 (6.9–15.7)	NR (2.7–25.7+)
Avelumab	JAVELIN Bladder 100 Phase III [32] June 2020	17%	21.4 (18.9–26.1) [33]	NR (10.5-NR)
**ADCs**				
Enfortumab vedotin ^4^	EV-201 Phase II [34] July 2019	44%	11.7 (9.1-NR)	7.6 (1.0–11.3)
Sacituzumab govitecan ^4,5^	IMMU-132 Phase I/II [35] February 2019	31%	18.9	12.6 (7.5–24.0)
**Small molecule**				
Erdafitinib ^4,6^	BLC2001 Phase II [36] July 2019	40%	13.8 (9.8-NR)	5.6 (4.2–7.2)

ICIs = Immune checkpoint inhibitor; ADCs = Antibody-drug conjugates; ORR = Objective response rate; OS = Median overall survival; DOR = Median duration of response; NR = Not reached. ^1^ For first-line treatment (cisplatin-ineligible). ^2^ For second-line treatment. ^3^ For first-line treatment (cisplatin-eligible). ^4^ Patient population included those who had previously received ICI therapy. ^5^ Not approved. Has been granted fast track designation by the FDA. ^6^ For patients with prespecified *FGFR* alterations.

**Table 2 ijms-21-07268-t002:** Treatment-related adverse events (AEs) from ICIs and targeted therapeutics.

AEs	Atezolizumab [21,23]	Nivolumab [29]	Pembrolizumab [26,27,28]	Avelumab [33]	Durvalumab [31]	Enfortumab Vedotin [34]	Sacituzumab Govitecan [66]	Erdafitinib [73]
% of any AE	69% ^1^/60% ^2^	69.3%	62% ^1^/64% ^2^	67%	60.7%	94%	NR ^3^	93%
% grade ≥3 AE	16%/15%	24.8%	16.9%/16%	8%	6.8%	54%	NR ^3^	46%
% discontinued due to AE	4%/6%	10%	6.8%/9.2%	6%	1.6%	12%	9%	13%
% treatment related deaths	0/1%	1.1%	0.8%/0.3%	0.6%	1%	0	0	0

^1^ Second-line setting. ^2^ First-line setting. ^3^ NR = not reported. Tagawa et al. [66], reports that the AE profile for sacituzumab govitecan in patients with bladder cancer was consistent with prior reports for breast cancer. Thus, the % of any AE and % grade ≥3 AE are from the reported clinical trial of sacituzumab govitecan in patients with metastatic triple-negative breast cancer, which was the basis for its approval for this cancer type [72], should be taken with caution.

**Table 3 ijms-21-07268-t003:** Pricing for ICIs and targeted therapeutics.

Drugs	$ ^1^/mg	Dose	$/Dose	CE ^2^
Pembrolizumab	$51.79	200 mg/3 weeks	$10,358	Difficult to justify ^4,5^
Nivolumab	$28.78	3 mg/kg/2 weeks	$7770 ^3^	No
Atezolizumab	$8.00	1200 mg/3 weeks	$9611	No ^4^/Likely ^5^
Durvalumab	$7.85	10 mg/kg/2 weeks	$7065 ^3^	No
Avelumab	$6.63	10 mg/kg/2 weeks	$5967	Yes
Enfortumab vedotin	$110	125 mg/kg/days 1, 8, 15 (28-day cycle)	$37,125 ^3^	Unknown
Sacituzumab govitecan	$11.20	10 mg/kg/days 1 and 8 (3-week cycle)	$20,120 ^3^	Unknown
Erdafitinib	$90	8 mg/day	$20,160 ^6^	Unknown

CE = Cost-effective. ^1^ Prices in U.S. currency. ^2^ At a $100,000 willing-to-pay threshold. ^3^ Calculated for 90 kg person. ^4^ For ‘after platinum-containing chemotherapy’. ^5^ For ‘cisplatin-ineligible’. ^6^ For 28-day supply.

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
