# Peer review of "Targeted Molecular Therapeutics for Bladder Cancer—A New Option beyond the Mixed Fortunes of Immune Checkpoint Inhibitors?"

_ijms, 2020, doi:10.3390/ijms21197268_

Round 1

Reviewer 1 Report

Solid contribution highlighting an emerging area of research and clinical impact. Although there are a few reviews on ADCs in urothelial carcinoma, this article is a comprehensive update, especially with the emphasis on clinical outcomes.

Author Response

Reviewer 1

Solid contribution highlighting an emerging area of research and clinical impact. Although there are a few reviews on ADCs in urothelial carcinoma, this article is a comprehensive update, especially with the emphasis on clinical outcomes.

Response. Thank you for appreciating the viewpoint from which I wrote this review. Immune checkpoint inhibitors (ICIs) are a mainstream exciting therapeutic approach for cancer. However, there acclaim overshadows the reality that they only work in a small proportion of patients. Thus the approval of 5 ICIs for metastatic bladder cancer, if viewed as ICIs are a great success, can be somewhat misleading. We aimed for an objective and comprehensive review that for the first time clearly demonstrates ICIs are somewhat of a mixed-fortune. As our research focuses on targeted therapeutics, specifically antibody-drug conjugates (ADCs), we describe the early evidence that suggests targeted agents may emerge as the next paradigm for the treatment of invasive or metastatic bladder cancer. 

Reviewer 2 Report

Generally this is an interesting review and relevant to current clinical practice though there a a few issues that should be addressed.

  1. The authors draw a number of conclusions from the described studies without adequately discussing limiting/confounding factors especially the fact that PD-L1 expression level is not a good biomarker for patient selection and in many cases is no longer required for access to immune checkpoint inhibitors. Further the differences in cut-offs between the different companion diagnostics are confounding and should be discussed.
  2. There is a significant bias in the writing towards favouring of targeted therapies that is not supported by the presented information. - there is very limited clinical trial data for the targeted therapies presented and limitations around follow-up time are not discussed. - different patient selection criteria between each of the studies is not discussed. - the authors surmise that 'the clinical effectiveness of ICIs is marginal and if it (is) worth paying a high price is justifiably questionable' There own analysis suggests that some of the therapies are indeed justified and that at this stage payment for the targeted therapies is not but this is not mentioned in this paragraph. - the very high rate of adverse events in response to targeted therapies is inappropriately down played, especially the significant grade 3 and higher AEs. This needs to be revised to appropriately reflect the data presented.
  3. Overall the background to the mutations being targeted and the mechanism of action of the targeted therapies is poorly described. Percentage of patients carrying the targetable mutations and likely to benefit should be described.
  4. on page 3 the statement is made "the mutation frequency in bladder cancer is high" - this should be clarified as referring to tumour mutation burden vs driver mutation frequency. TMB level should be compared to other cancers where ICI are approved.
  5. All acronyms should be defined at the first use.
  6. p4 line 143 'x months' data needs to be entered.
  7. Section 2.1.4 Claims are made that atezolizimab is not an effective monotherapy however PFS is not provided for the monotherapy group and OS was comparable to chemotherapy+placebo arm suggesting it was as effective at extending survival as the traditional therapy. While admittedly RR is lower in group B this is not the same as the therapy being ineffective and the conclusions should be revised.
  8. Line 296-7 'However, TMB and PD-L1 expression were not correlated as the there were only a small proportion to TMB patients' This is unclear and needs to be rephrased (?? of TMB patients, if so why?)
  9. Line 540-42 The sentence re Janssen comments on cost of their drug - this is unnecessary and does not add to the discussion of the topic and should be removed.
  10. The cost-effectiveness comparison in 6.2.7 is of limited relevance given the substantially different survival times in breast cancer patients.

Author Response

Generally, this is an interesting review and relevant to current clinical practice though there a a few issues that should be addressed.

Response. Thank you for appreciating the viewpoint from which we wrote this review. The other reviewer also appreciated our viewpoint. Immune checkpoint inhibitors (ICIs) are a mainstream exciting therapeutic approach for cancer. However, there acclaim overshadows the reality that they only work in a small proportion of patients. Thus the approval of 5 ICIs for metastatic bladder cancer, if viewed as ICIs are a great success, can be somewhat misleading. We aimed for an objective and comprehensive review that for the first time clearly demonstrates ICIs are somewhat of a mixed-fortune. As our research focuses on targeted therapeutics, specifically antibody-drug conjugates (ADCs), we describe the early evidence that suggests targeted agents may emerge as the next paradigm for the treatment of invasive or metastatic bladder cancer. We emphasize the word “may”, as you kindly point out, our bias is for targeted therapeutics. In general, we have toned down the rhetoric with respect to targeted therapeutics.

  1. The authors draw a number of conclusions from the described studies without adequately discussing limiting/confounding factors especially the fact that PD-L1 expression level is not a good biomarker for patient selection and in many cases is no longer required for access to immune checkpoint inhibitors. Further the differences in cut-offs between the different companion diagnostics are confounding and should be discussed.

Response. Thank you for this excellent point. Allow me to respond using the following points:

  • For each ICI we describe the assay for evaluating PD-L1 expression and how scoring was performed.
  • We describe important finding for patient responses based on PD-L1 expression subgroups.
  • We now include section 2.6 that concisely describes patient responses based on PD-L1 expression. Each ICI is described by bullet points in order for the reader to easily compare each ICI to one another.
  • We improved the description of PD-L1 as not a good biomarker in the Discussion.

We hope these changes are satisfactory.

  1. There is a significant bias in the writing towards favouring of targeted therapies that is not supported by the presented information. - there is very limited clinical trial data for the targeted therapies presented and limitations around follow-up time are not discussed. - different patient selection criteria between each of the studies is not discussed. - the authors surmise that 'the clinical effectiveness of ICIs is marginal and if it (is) worth paying a high price is justifiably questionable' There own analysis suggests that some of the therapies are indeed justified and that at this stage payment for the targeted therapies is not but this is not mentioned in this paragraph. - the very high rate of adverse events in response to targeted therapies is inappropriately down played, especially the significant grade 3 and higher AEs. This needs to be revised to appropriately reflect the data presented.

Response. Thank you very much for pointing out this bias. In general, we have toned down the rhetoric of favouring targeted therapeutics. We no longer claim they will take over ICIs but will be a new compliment for physicians to effectively treat patients with bladder cancer. Specific changes we have made are:

  • We have removed “more effective” form the main title.
  • Follow-up time: We agree that there is insufficient follow-up time for the targeted therapies to ascertain if they are really more effective than the ICIs. We mention this now in the Abstract and in the Discussion.
  • We revaluated the cost-effectiveness section based on revaluation of the clinical trial data. We have determined that only avelumab and atezolizumab (for cisplatin-inelgible patients only) are cost-effective. Thus, our original assessment holds, that with the exception of the above mentioned ICIs, these drugs are not cost-effective. We believe we provide a justified description that supports our overall viewpoint, that ICIs are a mixed fortune.
  • For Adverse Events, we agree and provide a note of caution in the Adverse Events section (lines 579-581, p. 14). We also mention this in the Discussion.

We hope these changes are satisfactory.

  1. Overall the background to the mutations being targeted and the mechanism of action of the targeted therapies is poorly described. Percentage of patients carrying the targetable mutations and likely to benefit should be described.

Response. Thank you for your points. We have provided concise descriptions of how the mechanisms of action for ADCs and erdafitnib. For the percentage of patients carrying targetable mutations, we believe you are referring to FGFR alterations. We state the introduction describing FGFR mutations, that FGFR mutations occur in >50% of all bladder cancer cases and that FGFR3 mutations occur in 60% of invasive bladder tumors. The BLC2001 trial, which evaluated the efficacy of erdafitinib, studies only patients with prespecified FGFR mutations. For the antigens for the two ADCs described in this review, we have revisited our descriptions and are satisfied that we adequately describe their expression in general in bladder cancer, and how expression levels were evaluated in the clinical trials.

  1. on page 3 the statement is made "the mutation frequency in bladder cancer is high" - this should be clarified as referring to tumour mutation burden vs driver mutation frequency. TMB level should be compared to other cancers where ICI are approved.

Response. We apologize for this mistake. Our intention is not to dive deep into the underlying patho-physiology of bladder cancer in the context of ICI therapy. We only aim to provide a brief statement on the underlying biology and refer the reader to an appropriate reference. However, TMB is important for the ICI nivolumab. We take this opportunity to describe in detail the associated findings between TMB and patient responses.

  1. All acronyms should be defined at the first use.

Response. Thank you for pointing this out. We have checked this and believe all acronyms are properly defined. 

  1. p4 line 143 'x months' data needs to be entered.

Response. Thank you for pointing this out. We have fixed this.

  1. Section 2.1.4 Claims are made that atezolizimab is not an effective monotherapy however PFS is not provided for the monotherapy group and OS was comparable to chemotherapy+placebo arm suggesting it was as effective at extending survival as the traditional therapy. While admittedly RR is lower in group B this is not the same as the therapy being ineffective and the conclusions should be revised.

Response. Thank you for pointing this out. We have revised the entire section describing the IMvigor130 trial. We hope it is satisfactory.

  1. Line 296-7 'However, TMB and PD-L1 expression were not correlated as the there were only a small proportion to TMB patients' This is unclear and needs to be rephrased (?? of TMB patients, if so why?)

Response. As mentioned in Point 4, we have rewritten this section.  

  1. Line 540-42 The sentence re Janssen comments on cost of their drug - this is unnecessary and does not add to the discussion of the topic and should be removed.

Response. Thank you for pointing this out. We have fixed this.

  1. The cost-effectiveness comparison in 6.2.7 is of limited relevance given the substantially different survival times in breast cancer patients.

Response. Thank you for pointing this out. We have removed this.